# Role of S100A8/A9 for Cytokine Secretion, Revealed in Neutrophils Derived from ER-Hoxb8 Progenitors

**DOI:** 10.3390/ijms22168845

**Published:** 2021-08-17

**Authors:** Yang Zhou, Justine Hann, Véronique Schenten, Sébastien Plançon, Jean-Luc Bueb, Fabrice Tolle, Sabrina Bréchard

**Affiliations:** Department of Life Sciences and Medicine, University of Luxembourg, 6 Avenue du Swing, L-4367 Belvaux, Luxembourg; yang.zhou@uni.lu (Y.Z.); justine.hann@uni.lu (J.H.); veronique.schenten@uni.lu (V.S.); sebastien.plancon@uni.lu (S.P.); jean-luc.bueb@uni.lu (J.-L.B.); fabrice.tolle@uni.lu (F.T.)

**Keywords:** Hoxb8 neutrophils, cytokine secretion, S100A8/A9, degranulation

## Abstract

S100A9, a Ca^2+^-binding protein, is tightly associated to neutrophil pro-inflammatory functions when forming a heterodimer with its S100A8 partner. Upon secretion into the extracellular environment, these proteins behave like damage-associated molecular pattern molecules, which actively participate in the amplification of the inflammation process by recruitment and activation of pro-inflammatory cells. Intracellular functions have also been attributed to the S100A8/A9 complex, notably its ability to regulate nicotinamide adenine dinucleotide phosphate (NADPH) oxidase activation. However, the complete functional spectrum of S100A8/A9 at the intracellular level is far from being understood. In this context, we here investigated the possibility that the absence of intracellular S100A8/A9 is involved in cytokine secretion. To overcome the difficulty of genetically modifying neutrophils, we used murine neutrophils derived from wild-type and S100A9^−/−^ Hoxb8 immortalized myeloid progenitors. After confirming that differentiated Hoxb8 neutrophil-like cells are a suitable model to study neutrophil functions, our data show that absence of S100A8/A9 led to a dysregulation of cytokine secretion after lipopolysaccharide (LPS) stimulation. Furthermore, we demonstrate that S100A8/A9-induced cytokine secretion was regulated by the nuclear factor kappa B (NF-κB) pathway. These results were confirmed in human differentiated HL-60 cells, in which S100A9 was inhibited by shRNAs. Finally, our results indicate that the degranulation process could be involved in the regulation of cytokine secretion by S100A8/A9.

## 1. Introduction

S100A9, together with its partner S100A8, are cytosolic Ca^2+^-binding proteins highly expressed in neutrophils, which can operate as homo- or heterodimers [1,2]. An impediment in the formation of tetramers has been associated with a deficiency of intracellular S100A8/A9 functions highlighting the fact that the heterotetramers, which are specifically formed during an increase of Ca^2+^ concentration [3], represent the privileged form for the activity of S100A9 [4]. It has later been reported that due to their conformational state after Ca^2+^-binding, the preferential structure for human S100A8 and S100A9 is the heterodimer S100A8/A9 [5].

S100A8 and S100A9 are secreted in the extracellular environment, where they act as danger associated molecular pattern (DAMP) molecules and they exert their effects predominantly through the binding to Toll-like receptor 4 (TLR4) [6,7] and possibly via the receptor for advanced glycation end products (RAGE) [8].

Secreted S100A8 and S100A9 have a chemotactic role [9] and consequently, can increase the migration of inflammatory cells, which are responsible for the secretion of pro-inflammatory cytokines [10]. S100A8/A9 can also decrease the adaptive T cell immune response by preventing the differentiation and development of dendritic cells [11]. Moreover, S100A9 is able to promote IL-8 secretion and a degranulation process by human neutrophils [12] as well as pro-inflammatory cytokine production by monocytes and macrophages via the nuclear factor kappa B (NF-κB) and p38 mitogen-activated protein kinase (MAPK) pathways [13].

At the intracellular level, S100A8/A9 complexes regulate nicotinamide adenine dinucleotide phosphate (NADPH) oxidase activation through (i) direct interaction with cytochrome *b*_558_ [14] and binding of S100A8 with p67^phox^ as well as Rac2 and (ii) by binding to arachidonic acid facilitating the transfer of arachidonic acid to the NADPH oxidase complex [15]. However, the impact of intracellular S100A8/A9 on the regulation of pro-inflammatory functions of neutrophils remains elusive. This is probably due to the fact that purified neutrophils have a short life span [16] and are thus difficult to maintain in culture. In this sense, human neutrophils represent one of the last cell types which cannot be genetically manipulated. A promising surrogate neutrophil model is based on the immortalization of a murine myeloid progenitor lineage by overexpression of the oncoprotein Hoxb8, which can facilitate studies to investigate neutrophil biological functions in more detail [17].

The ER-Hoxb8 (estrogen regulated-homeobox b8) cell line carries a fusion of Hoxb8, a member of the Homeobox gene family, with an estrogen response element in the genome of myeloid precursor cells. ER-Hoxb8 cells can be differentiated toward neutrophils upon estrogen starvation and thus Hoxb8 protein silencing. The use of ER-Hoxb8 cells makes it possible to conditionally differentiate neutrophils derived from genetically modified mice, especially from S100A9 knock-out mice. Previous studies reported the ability of Hoxb8-derived neutrophil-like cells to recapitulate key functions of neutrophils including phagocytosis and production of reactive oxygen species (ROS) stimulated by yeast particles [17,18,19].

Using ER-Hoxb8 cells from S100A9 wild-type and knock-out mice as well as human differentiated HL-60 cells infected with lentiviral S100A9-shRNAs, we demonstrated a novel role for S100A8/A9, which is the regulation of cytokine secretion through NF-κB activation.

## 2. Results

### 2.1. Characterization of S100A9^−/−^ Hoxb8 Neutrophil-like Cells

First of all, we determined whether S100A9^−/−^ ER-Hoxb8 cells could constitute a relevant tool to study the role of S100A8/A9 in pro-inflammatory functions of primary neutrophils since these cells cannot be genetically modified and therefore no functional experiments can be performed after knock out of a protein of interest. To achieve this aim, we controlled whether S100A9^−/−^ cells were able to differentiate into neutrophils upon five days of estrogen starvation similarly to WT Hoxb8 cells. After 5 days of differentiation, WT and S100A9^−/−^ Hoxb8 cells presented the typical polymorphic nucleus of neutrophils compared to their respective progenitors (Figure 1a) while the viability of differentiated WT and S100A9^−/−^ Hoxb8 cells was not affected throughout the differentiation process (Figure 1b). To evaluate the state of differentiation of Hoxb8 cells into neutrophils, progenitor and granulocytic marker expression was measured by flow cytometry. As expected, a down-regulation of the progenitor marker CD71 was observed upon differentiation while CD11b and Ly6C/6G were up-regulated in WT Hoxb8 cells. The expression level of F4/80 remained unchanged before and after differentiation (Figure 1c).

The rate of differentiation of WT ER-Hoxb8 and S100A9^−/−^ ER-Hoxb8 into neutrophil-like cells was similar after removal of estrogen for five days underlining the fact that the absence of S100A8/A9 had no influence on the process of differentiation (Figure 1c).

Since the expression of S100A8 and S100A9 has been shown to be induced during the differentiation process of HL-60 cells [20], the gene and protein expression of S100A8 and S100A9 have been measured in differentiated Hoxb8 cells. S100a8 and S100a9 mRNAs were highly expressed after 5 days of estrogen removal in differentiated WT Hoxb8 cells (Figure 1d). In the same way, S100A8 and S100A9 proteins were detected in the cytosol of the differentiated WT Hoxb8 cells but not in those of the progenitors (Figure 1d). For S100A9^−/−^ Hoxb8 cells, as expected, no expression of S100A9 at mRNA and protein level was detected. S100a8 mRNA expression was found at a similar level between WT and S100A9^−/−^ Hoxb8 cells, whereas S100A8 protein was largely decreased in differentiated S100A9^−/−^cells (Figure 1d). The absence of S100A8 protein [21,22] has been previously proposed to be provoked by an inefficient translation of S100A8 mRNA but the predominant hypothesis is that the absence of S100A9 protein leads to an instability of S100A8 protein [22].

Thus, S100A9^−/−^ Hoxb8 cells were also deficient in S100A8/A9, consistent with the fact that S100A8 and S100A9 homodimers do not exist in vivo [23].

Finally, because it is well-established that S100A8/A9 plays a role in the regulation of the neutrophil NADPH oxidase [26,27,28], the impact of the absence of S100A8/A9 on ROS production was determined in Hoxb8 cells differentiated into neutrophil-like cells. Differentiated S100A9^−/−^ Hoxb8 cells expressed all subunits of the NADPH oxidase. The protein bands matched with the molecular weights predicted in mice except for p67^phox^, which was smaller than the expected (Figure 2a).

As previously described in neutrophil-like differentiated HL-60 cells [28], ROS production was decreased in S100A9^−/−^ Hoxb8 cells in response to fMLF. In contrast, PMA-induced ROS production was not impaired by the absence of S100A9 (Figure 2b) confirming that S100A9^−/−^ cells have a functional NADPH oxidase system. Finally, the absence of S100A8/A9 led to a reduced ROS production after LPS stimulation of S100A9^−/−^ Hoxb8 cells when compared to WT Hoxb8 cells (Figure 2b). To our knowledge, this is the first report on the involvement of S100A8/A9 in LPS-mediated regulation of NADPH oxidase activation.

Altogether, our data show that S100A9^−/−^ Hoxb8 cells represent a suitable model to determine the role of S100A8/A9 in neutrophil cytokine secretion.

### 2.2. S100A8/A9 Modulates Cytokine Secretion

To study the impact of the absence of intracellular S100A8/A9 on the release of cytokine secretion, Hoxb8 neutrophil-like cells were stimulated with LPS, which is known to induce the secretion of a broad range of cytokines. The mRNA expression and secretion of several cytokines described to be induced by LPS and relevant for neutrophil-related pro-inflammatory diseases, has been investigated in differentiated WT Hoxb8 cells. The mRNA expression of *Ccl2*, *Ccl3*, *Ccl4*, *Il-1α*, *Il-1β*, *Il-6*, *Tnf-α* and *Cxcl2* was increased upon LPS stimulation (Figure 3). For all cytokines except *Ccl2*, expression was increased at 2 h of stimulation and remained elevated for up to 12 h of stimulation. The expression of *Ccl2* started at 4 h of stimulation and increased over time.

Although the expression of all cytokines was up-regulated upon LPS stimulation, the magnitude of expression varied between different cytokines. The correlation between levels of protein secretion and induced RNA expression varied amongst different cytokines. Indeed, IL-1α and IL-1β secretion was hardly induced in LPS-stimulated cells, which matched the rather low levels of mRNA that were detected for the different time points. tumor necrosis factor alpha (TNF-α) and C-C motif chemokine ligand (CCL)-3 were secreted after 2 h of stimulation whereas interleukin (IL)-6 and CCL4 release peaked after 4 h of LPS treatment. C-X-C motif chemokine ligand 2 (CXCL2) and CCL2 were secreted only at the 12 h stimulation point. CCL3, CCL4, IL-6, TNF-α and CXCL2 were highly secreted while the level of CCL2 released was moderate (Figure 4).

Since WT Hoxb8 cells have the ability to modulate mRNA expression and cytokine secretion upon LPS stimulation, we investigated whether the absence of intracellular S100A8/A9 altered these two processes. S100A8/A9 deficiency triggered a general dysregulation of cytokine mRNA expression (Figure 5) compared to WT Hoxb8 cells (Figure 3).

The effect of S100A8/A9 on cytokine mRNA expression levels correlated with the magnitude of secretion for some cytokines but not for all. Although CCL2, IL-6 and CXCL2 secretion was reduced in absence of intracellular S100A8/A9, the release of CCL3, CCL4 and TNF-α was up-regulated in S100A9^−/−^ cells upon 6 to 12 h of LPS stimulation (Figure 6). It is important to note that IL-1α and IL-1β were not secreted upon LPS stimulation in differentiated S100A9^−/−^ Hoxb8 cells as previously observed in differentiated WT Hoxb8 cells.

### 2.3. Is Cytokine Secretion Mediated by the Process of Degranulation?

The mechanisms associated with cytokine secretion by neutrophils are largely unknown but the idea that degranulation could be involved in the release of cytokines is increasingly recognized [29]. Therefore, we set out to investigate whether S100A8/A9 could influence the cytokine secretion via a regulatory effect on the process of degranulation. For this purpose, secretion of soluble mediators described to be characteristic for the different granule types (MPO for azurophil granules, lipocalin for specific granules, MMP-9 for gelatinase granules and albumin for secretory vesicles) was measured by ELISA following LPS stimulation of differentiated WT and S100A9^−/−^ Hoxb8 cells. Unfortunately, the high basal levels of soluble granule matrix proteins in the non-stimulated cells appeared to disguise the effect of LPS on their secretion in our experimental conditions (data not shown). Therefore, the impact of the absence of intracellular S100A8/A9 on degranulation has been determined by evaluating the translocation of CD markers associated with each type of granules. While translocation of CD11b and CD14 at the plasma membrane was observed in differentiated Hoxb8 cells, it must be noted that in our hands, differentiated Hoxb8 cells were not able to release azurophil granules (Figure 7a) as previously described [30,31]. Similarly, no significant translocation of CD18 was observed in our experimental conditions highlighting the fact that no specific granules were released upon LPS stimulation (Figure 7a). In accordance with these data, no difference on the translocation of CD63 and CD18 was observed between differentiated WT and S100A9^−/−^ Hoxb8 cells. Moreover, the relocation of CD11b at the plasma membrane upon LPS stimulation was unchanged in the absence of S100A9 (Figure 7b). Only the translocation of CD14 was clearly inhibited in S100A9 deficient cells at 12 h of LPS stimulation. Although our data seem to indicate that S100A9 could be involved in the degranulation of secretory vesicles, the incomplete granule profile in Hoxb8 cells is a limitation to their use for degranulation experiments and cannot be replaced by assays with primary neutrophils as already reported [31].

### 2.4. The Effects of S100A9 on Cytokine Secretion Are Mediated by NF-κB

Besides the assumption that S100A8/A9 might influence cytokine secretion through the process of degranulation, another possibility is that S100A8/A9 alters the level of cytokine secretion through the NF-κB pathway. Indeed, Simard et al. [32], showed that IL-6 and IL-8 secretion was reduced after NF-κB inhibition. In our experiments, we blocked NF-κB signaling with BAY 11-7082, an inhibitor of NF-κB inhibitor (IκΒα) phosphorylation, prior to stimulation of differentiated HoxB8 cells with LPS. Our data showed that treatment with BAY 11-7082 strongly reduced cytokine secretion. However, no differential effect of BAY 11-7082 was observed on LPS-induced cytokine secretion between differentiated WT and S100A9^−/−^ Hoxb8 cells (Figure 8).

The molecular mechanisms leading to cytokine secretion could differ between mice and human [33,34]. Therefore, we used human differentiated HL-60 cells in which S100A9 expression was downregulated by shRNAs to examine whether cytokine secretion can be regulated by S100A9 in an NF-κB-dependent manner. As expected, both S100A8 and S100A9 expression was inhibited by S100A9-shRNAs (Figure 9a) and ROS production almost completely abolished in S100A9-shRNA cells (Figure 9a). Moreover, inhibition of S100A9 in differentiated HL-60 cells was also able to affect cytokine secretion upon LPS stimulation. In this regard, CCL3, IL-6 and IL-8 secretion was largely decreased in absence of S100A8/A9 while CCL2 secretion tended towards a reduction. However, none of the tested cytokines was found to have increased secretion in S100A9-shRNA cells compared to control cells (Figure 9b). Noteworthy, the basal level of cytokine secretion in control and S100A9-shRNA was identical (data not shown). As observed for Hoxb8 cells, the blockade of NF-κB signaling with BAY 11-7082 strongly reduced cytokine secretion, with no differential effect between differentiated control and S100A9-shRNA HL-60 cells (Figure 9c).

## 3. Discussion

The importance of S100A8/A9 as a biomarker for inflammatory diseases such as rheumatoid arthritis [35] as well as its multiple extracellular functions have been described in detail over the past years [36]. However, and even though S100A9 together with S100A8, represents one of the most abundant protein complexes found in the cytosol of neutrophils, a knowledge gap remains between its established involvement in the regulation of neutrophil pro-inflammatory functions and its potential contribution to cytokine secretion.

Indeed, neutrophils being terminally differentiated with a short lifespan, neither expandable in culture nor genetically modifiable, makes them ill-suited for long-term experiments necessary for the characterization of functional roles of S100A8/A9.

The recently developed ER-Hoxb8 system constitutes a promising tool for studying the ability of intracellular S100A8/A9 to modulate neutrophil functions since immortalization of myeloid progenitors extracted from S100A9^−/−^ or WT C57/BL6 mice by the Hoxb8 oncoprotein can be easily obtained. Using this cell system, our results clearly demonstrate that intracellular S100A8/A9 is able to regulate cytokine secretion.

Although mechanisms associated with such processes are not yet elucidated, it cannot be excluded that S100A8/A9 can regulate degranulation, which could be important for the process of cytokine secretion. Indeed, in the last few years a consensus emerged in favor of a release of pre-formed cytokines by exocytosis upon ligand receptor signaling. In this respect, pre-formed cytokines could be stored in neutrophil granules awaiting their release into the extracellular space [37,38,39,40,41]. In accordance with this assumption, our data provide evidence for a role of S100A8/A9 in the secretion of all tested cytokines (CCL2, CCL3, CCL4, IL-6, TNF-α and CXCL2) except IL-1β. Indeed, IL-1β secretion is independent of the process of the regulated exocytosis and occurs through plasma membrane transporters [42]. Furthermore, our data show that the absence of S100A8/A9 alters the mobilization of secretory vesicles, strengthening the fact that S100A8/A9-modulated cytokine secretion could be dependent on the degranulation process.

In fact, S100A8/A9 is involved in the formation of microtubules in the cytosol, which allow the translocation of granules to the plasma membrane [4,43]. Thus, S100A8/A9 deficiency may disrupt the linkage between cytoskeleton and microtubules leading to the perturbation of the degranulation process.

Moreover, several Ca^2+^-binding proteins, including synaptotagmin II, annexins and Munc13-4 have been described to promote the membrane fusion in neutrophils [44,45,46]. In this context, as a Ca^2+^-binding protein, S100A8/A9 may translocate from the cytosol to the plasma membrane and regulate the fusion between secretory granules and the membrane.

As supported by our results, S100A8/A9 is modulating cytokine secretion through the degranulation process. However, while our data provide first evidence that S100A8/A9 could be involved in the release of secretory vesicle contents upon LPS stimulation, we however cannot exclude the possibility that S100A8/A9 is also involved in the regulation of the release of other granule types. It is important to note that no translocation of CD63 and CD18 to the cell surface was observed in our experimental conditions underlining the fact that differentiated Hoxb8 cells were able to express primary and secondary granules. In disagreement with our results, it has been reported that Hoxb8 neutrophil-like cells released secondary granules in addition of tertiary granules, however only upon FcγR stimulation [30]. Indeed, it appears that the mobilization of granules is dependent on the type and concentration of the agonist [47]. Moreover, in the few studies performed on Hoxb8 cells, proteins found in each type of granule have been reported to be lowly expressed in comparison to those present in primary mouse neutrophils. Altogether, these data show that differentiated Hoxb8 cells are not yet a perfect model to study the mechanisms surrounding the process of degranulation.

Another point we have addressed in our study, is the link between S100A8/A9 and NF-κB signaling pathway. Indeed, our results show that BAY 11-7082, an inhibitor of IκΒα phosphorylation, is able to largely inhibit cytokine secretion. However, our data do not support clear conclusions on the role of S100A9 in the transcriptional response underlying cytokine synthesis. The ability of S100A8/A9 to bind Ca^2+^ could be responsible for its effect observed on cytokine RNA expression through the inhibition of NF-κB activation. Indeed, S100A8/A9 may sequester Ca^2+^, which has been described to be involved in Iκκα/β phosphorylation and NF-κB-activation in neutrophils [48] by preventing cytokine synthesis via inhibition of NF-κB activation.

The axis S100A8/A9-NF-κB in the regulation of cytokine transcription may also involve ROS. On the one hand, ROS has been reported to directly or synergistically up-regulate NF-κB activation [49], and on the other hand, oxidation of NF-κB by ROS triggers a decrease of DNA binding activity in the nucleus [50]. Moreover, it has been well-established that intracellular S100A8/A9 is a primordial regulator of the NADPH oxidase activation [26,51].

The exact mechanisms of S100A9 involvement in the regulation of cytokine expression remain to be determined. Understanding how cytokine production is regulated from the perspective of gene regulation is an issue of major concern since cytokine dysregulation is associated with a battery of diseases autoimmune disorders, chronic inflammatory and cancer [52,53,54,55]. In this context, S100A9-dependent regulatory pathways could represent a major target to modulate cytokine expression and dampen an aberrant cytokine production or privilege the production of anti-inflammatory cytokines for controlling inflammation. Thus, the ultimate goal would be to regulate immune/autoimmune responses and at the same time prevent collateral damages that could potentially induce a pathological situation related to immunodeficiency or exacerbated inflammation.

Another important point and as mentioned before, is the fact that the molecular mechanisms leading to cytokine secretion may not be identical in mice and humans. It is well described that major differences exist concerning the type of cytokines released by the two species. For example, IL-10 has been found to be secreted by murine but not by human neutrophils [56]. Furthermore, IL-6 expression by human neutrophils is still being debated. Therefore, it is imperative to also validate the involvement of S100A9 in the regulation of cytokine secretion in a model of human neutrophils. As primary neutrophils are genetically not modifiable, we have used HL-60 cells, which are differentiated human promyelocytic leukemia cells. We downregulated S100A9 by shRNA and stimulated such treated HL-60 cells with LPS. As for Hoxb8 cells, CCL2, CCL3 and IL-6 was secreted whereas no secretion of TNF-α was detected in HL-60 cells. The absence of S100A9, as observed in mouse cell lines, affected the secretion of these cytokines in the human cell line. The secretion of IL-8, which is not produced by mice, was also decreased in S100A9-shRNA HL-60 cells. Moreover, NF-κB was also associated with the decrease of cytokine secretion in differentiated HL-60 cells underlining the fact that Hoxb8 can constitute a model adapted to study the functional mechanisms connected to S100A9 proteins. However, the fact that CCL3 and CCL4 secretion were increased and not decreased in S100A9^−/−^ Hoxb8 cells, seem to indicate a difference in regulation between mice and human cells and require further investigation.

Besides the absence of certain types of granules, differentiated Hoxb8 cells also have other limitations. Indeed, the process leading to the differentiation of Hoxb8 cells still needs to be improved. However, the rate of differentiation of Hoxb8 cells, obtained from our experimental conditions, was in line with those previously reported [17,57,58]. Saul et al. [31] have even recently succeeded reaching a rate of differentiation of approximately 90% by adding IFN-γ and TNF-α during the differentiation process. However, a further improvement of the percentage of differentiation and exclusion of contaminating cells (e.g., monocytes) are still required to reach a rate of ≥99% of neutrophils.

Even with its present limitations, the Hoxb8-system currently constitutes the only cellular model from which knockout genes and unlimited source of primary neutrophils can be easily obtained. In that context and for the first time we were able to demonstrate that S100A8/A9 is a regulator of cytokine secretion through NF-κB-activation in differentiated Hoxb8 cells and that the process of degranulation might be altered by S100A8/A9. In addition, further experiments are now required to determine more precisely the role of intracellular versus extracellular S100A8/A9 in the regulation of cytokine secretion and define whether intracellular S100A8/A9 are similarly involved in signalling pathways activated by S100A8/A9 and LPS.

## 4. Materials and Methods

### 4.1. Cell Culture and Differentiation of ER-Hoxb8 Progenitors to Neutrophil-like Cells

The human promyelocytic leukemia HL-60 cell line (ATTC, #CCL-240) was cultured in RPMI-1640 supplemented with 2 mM L-glutamine, 10% complement heat-inactivated fetal bovine serum (FBS), 100 μg/mL streptomycin and 100 U/mL penicillin. Cells were incubated at 37 °C, 5% CO_2_ and differentiated into neutrophil-like cells (dHL-60) by 1.3% *v/v* dimethylsulfoxide (DMSO) for 4.5 days [59].

Hoxb8 neutrophil progenitors were originally established from bone marrow of wild type (WT) or S100A9^−/−^ C57/BL6 mice and immortalized by ER-Hoxb8 oncoproteins. The cell lines were kindly provided by Pr. Thomas Vogl (University of Münster, Germany) and the University of California, San Diego (Pr. Mark P. Kamps).

The ER-Hoxb8 progenitors were seeded at 0.2 × 10^6^ cells per mL in OptiMEM medium (Life Technologies, Gent, Belgium) supplemented with 10% heat-inactivated fetal bovine serum (Sigma-Aldrich, Bornem, Belgium), 2 mM L-glutamine (Life Technologies), a mix of 100 µg/mL streptomycin and 100 U/mL penicillin (Life Technologies), 1% *v/v* of culture supernatant from stem cell factor (SCF)-producing CHO cells and 1 µM β-Estradiol (Sigma-Aldrich) as described earlier [17]. The cells were kept at 37 °C in a humidified atmosphere with 5% CO_2_ and sub-cultivated twice a week.

Differentiation of ER-Hoxb8 progenitors (0.6 × 10^6^ cells per mL) to neutrophil-like cells was induced by removal of β-estradiol in the presence of SCF (1% *v/v* of culture supernatant) in RPMI-1640 medium (Life Technologies) supplemented with 10% heat-inactivated fetal bovine serum, 2 mM L-glutamine, 100 µg/mL streptomycin and 100 U/mL penicillin. The medium without β-estradiol was replaced after two days of differentiation.

### 4.2. May-Grünwald Giemsa Staining

Hoxb8-like cells were spread on microscope slides and directly fixed/stained in pure May-Grünwald solution (Merck-Millipore, Overijse, Belgium) for 10 min. Then, the slides were stained in Giemsa solution (Merck-Millipore; 10% *v/v* in H_2_O) for 30 min followed by an incubation in pure water until a light blue coloration was observed. Analyses were made by microscopy (Leica, Wetzlar, Germany) using an oil immersion objective (numerical aperture of 1.4 and magnification of 63).

### 4.3. Cell Viability

Cell viability was assessed by using 7-amino actinomycin D (7-AAD; Milteny Biotec, Leiden, The Netherlands). Briefly, 0.5 × 10^6^ cells were washed twice with PBS and resuspended in FACS Buffer (137 mM NaCl, 2.6 mM KCl, 8 mM Na_2_HPO_4_, 1.8 mM KH_2_PO_4_, 0.15% *w/v* BSA, 0.05% *w/v* NaN_3_, pH 7.4) containing 2.5 µg/mL 7-AAD. After 30 min at 4 °C protected from the light, 7-AAD fluorescence (excitation wavelength at 488 nm and a maximum emission wavelength at 647 nm) was determined using the PerCP-Cy5.5 filter of the BD FACs Canto II™ instrument (BD Biosciences, Erembodegem, Belgium). Living cells were determined as percentage of 7-AAD-negatively labelled cells on the entire cell population.

### 4.4. Analysis of Surface Marker Expression by Flow Cytometry

Expression of F4/80, Ly-6C/6G, CD11b and CD71 was determined by flow cytometry. Cells (0.5 × 10^6^) were washed with PBS, suspended in FACS buffer and blocked during 15 min at 4 °C with 10% of normal mouse serum (ThermoFisher Scientific, Erembodegem, Belgium). Then, the cells were stained with the following fluorochrome-conjugated rat anti-mouse antibodies: anti-F4/80-APC (ThermoFisher Scientific, clone BM8), anti-Ly6C/6G-eFluor^®^450 (ThermoFisher Scientific, clone RB6-8C5), anti-CD11b-FITC (ThermoFisher Scientific, clone M1/70) and anti-CD71-PE (ThermoFisher Scientific, clone R17 217.1.4). Cell samples were kept unstained or incubated with the following isotype control: rat IgG2b-APC (ThermoFisher Scientific, clone eBr2A), rat IgG2b-FITC (ThermoFisher Scientific, clone eB149/10H5), rat IgG2a-PE (ThermoFisher Scientific, clone eBr2a), and rat IgG2b-eFluor^®^450 (ThermoFisher Scientific, clone eB149/10H5). Cells were incubated with antibody cocktails for 30 min at 4 °C in the presence of 7-AAD. Surface marker expression was determined on a BD FACS Canto II™ flow cytometer (BD Biosciences).

### 4.5. Total RNA Extraction and Quantitative RT-PCR

Total RNA was isolated using TRIzol (Life Technologies) which was added to the cell pellet (10^7^ cells/condition). Then, RNA was separated from DNA and proteins by addition of chloroform. The aqueous phase was removed and RNA precipitation was performed by addition of an equal volume of 100% isopropanol followed by 10 min incubation at room temperature. Precipitated RNA was pelleted through centrifugation (14,000× *g*, 10 min) and washed twice with ethanol 70% before elution in RNase-free water.

cDNA was prepared from 0.5 µg RNA using 0.2 μg random hexamers, 20U RNAsin Ribonuclease inhibitor (Promega, Madison, WI, USA), 0.5 mM dNTP, and 40 U GoScript reverse transcriptase (Promega). Reverse transcription was performed as follows: 5 min at 25 °C, 60 min at 42 °C, and 10 min at 70 °C. Reverse transcription was performed in 9700 GeneAmp thermocycler (ThermoFisher Scientific). PCR primers for target and reference genes were designed according to published sequences in GenBank using Primer3 online software. qPCR was performed using the SYBR^®^ Select master Mix (ThermoFischer Scientific) in a QuantStudio 12K Flex real-time PCR machine (ThermoFischer Scientific). The cycling protocol was as follows: 3 min at 50 °C and 3 min at 95 °C followed by 40 cycles of 3 s at 95 °C and 30 s at 60 °C. The relative quantification of the different mRNAs was normalized using the three reference genes (*Actβ*, *Gusβ*, *B2m*) according to the Vandesompele method, based on the geometric averaging of multiple internal controls [25]. Primer sequences used for qRT-PCRs are listed in the Table 1.

### 4.6. Knock-Down of S100A9 in HL-60 Cell by shRNA Interference

The viral delivery to knock-down the S100A9 expression was performed using Lenti-X Packaging Single Shots (VSV-G) kit from Takara Bio (Clontech, 631276; Saint-Germain-en-Laye, France). Lentiviral plasmids for shRNA expression were purchased from Dharmacon (Lafayette, CO, USA). Lentivirus was produced by transfecting HEK293T cells with the plasmids at 80–90% confluence in DMEM (high glucose, supplemented with 10% FBS, 2 mM L-glutamine, 100 µg/mL streptomycin and 100 units/mL penicillin) in fibronectin coated Petri dish according to the manufacturer’s instructions. Fresh media was added to the cells to adjust the volume to 15 mL. Virus containing supernatant was harvested 48 h and 72 h after infection, pre-cleaned with 3000× *g* centrifugation and a 0.45-μm filtration (Merck-Millipore). The virus containing supernatant was further overlaid on a sucrose-containing buffer (10% sucrose p/v, 50 mM Tris-HCl, pH 7.4, 100 mM NaCl, 0.5 mM EDTA) and centrifuged at 10,000× *g* for 4 h under 4 °C [60]. The supernatant was then removed and virus was resuspended with Hank’s salt solution (Sigma-Aldrich, H6648) at 4 °C overnight in dark and stored in 10^7^ IFU fraction at −80 °C the next day after titration. For HL-60 infection, 100 µL of virus (~10^7^ IFU) supplemented with 6 µg/mL of DEAE-dextran was preloaded into each well of a 24-well plates pre-coated with retronectin (20 µg/mL) in ACD-A (citrate dextrose form A) solution and then centrifuged at 2000× *g* for 2 h at 32 °C [61]. Then, 10^6^ HL-60 cells were added to each virus-containing well and cultured at 37 °C with 5% CO_2_. Six hours after, the cells/virus suspension was diluted 5 times with media in order to stop the infection. Two days after, the 14 days’ cell selection started by adding 0.3 µg/mL puromycin to fresh media exchanger three times a week.

### 4.7. Western Blot

WT or S100A9^−/−^ ER-Hoxb8 cells were lysed in a fresh Triton lysis buffer (50 mM Tris pH 8.0, 150 mM NaCl, 1% Triton, 5% glycerol, SigmaFast protease inhibitor cocktail and phosphatase inhibitor cocktails 2 and 3 from Sigma-Aldrich). The protein concentration was determined by using Pierce BCA Protein Assay Kit (ThemoFisher Scientific). Loading buffer (Tris 63 mM pH 6.8; 2% SDS; 10% Glycerol; 1% β-mercaptoethanol) was added to 50 µg of the different protein samples and then heated for 10 min at 96 °C. Protein samples were run on a Tris-Tricine gel (10% acrylamide) and then electrotransferred to a PVDF membrane (0.2 µm, Merck-Millipore). Immunodetection of S100A8 was performed using the monoclonal rat anti-mouse S100A8 antibody, clone ABM4A69 (Abcam, Cambridge, UK) and monoclonal anti-human S100A8, clone EPR3554 (Abcam, ab92331). Immunodetection of S100A9 was assessed by using monoclonal rat anti-mouse S100A9 antibody (clone 372510; Bio-Techne, Abingdon Oxon, UK) and monoclonal anti-human S100A9, clone B-5 (Santa Cruz, sc-53186). NADPH oxidase proteins (p22^phox^, p40^phox^, p47^phox^, p67^phox^ and gp91^phox^) were detected using the polyclonal rabbit anti- p22^phox^ antibody (Santa Cruz Biotechnology, Heidelberg, Germany), the monoclonal rabbit anti-p40^phox^ antibody, clone EP2142Y (Abcam), the polyclonal rabbit anti-mouse p47^phox^ and anti-mouse p67^phox^ (Merck-Millipore), and the monoclonal rabbit anti-gp91^phox^ (clone EPR6991; Abcam). Protein loading quantity was determined using the monoclonal mouse anti-actin, (clone C4; Merck-Millipore). Finally, goat anti-rat coupled with IRDye^®^ 800CW (for S100A8/A9) and donkey anti-mouse or rabbit coupled with IRDye^®^ 800CW or IRDye^®^ 680RD (for NADPH oxidase protein and actin) were used as fluorescent secondary antibodies for protein detection using the Odyssey Infrared Imaging system (LI-COR Bioscience, Lincoln, NE, USA).

### 4.8. ROS Production

Differentiated ER-Hoxb8 cells (0.5 × 10^6^) were resuspended in a physiological salt solution (PSS; 115 mM NaCl, 5 mM KCl, 1 mM KH_2_PO_4_, 10 mM D-Glucose, 1 mM MgSO_4_, 1.25 mM CaCl_2_ and 25 mM HEPES, pH 7.4) containing 20 µM luminol (Sigma-Aldrich) and 10 U/mL horseradish peroxidase (Sigma-Aldrich). The NADPH oxidase activity was measured overtime by chemiluminescence after addition of fMLF (Sigma-Aldrich), PMA (Sigma-Aldrich) or LPS (from *E. coli*, Sigma-Aldrich). Luminescence photon emission was recorded every 30 s for 4 h at 37 °C in a Clariostar plate reader (CLARIOstar, BMG LABTECH, Ortenberg, Germany).

Differentiated HL-60 cells (2 × 10^6^) were resuspended, for 10 min at 37 °C, in PSS containing 30 µM Amplex Red (Sigma-Aldrich) and 1 U/mL horseradish peroxidase (Sigma-Aldrich). The NADPH oxidase activity was measured overtime by fluorescence after addition of fMLF with a Quantamaster spectrofluorimeter QM-8/2003 (Photon Technology International, Inc., Lawrenceville, NJ, USA).

### 4.9. Measurement of Cytokine Secretion by Cytometric Bead Array (CBA) and ELISA

Fresh supernatants of differentiated ER-Hoxb8/HL-60 cells (2 × 10^6^ cells/mL), stimulated by LPS in RPMI-1640 (for ER-Hoxb8)/PSS (for HL-60 cells), were collected for subsequent quantitative measurement of cytokine secretion by CBA following the manufacturer’s instructions (BD Biosciences). Briefly, the multiplex standard curve composed of mixed cytokine standards was set-up by serial dilutions. Selected capture beads were mixed and added to either the supernatants or the mixed standard cocktail. The following anti-mouse beads were used: CCL2 (MCP-1, bead B7), CCL3 (MIP-1α, bead C7), CCL4 (MIP-1β, bead C9), IL-1α (bead E4), and TNF-α (bead C8). For human cytokine secretion, CCL2 (MCP-1, bead D8), CCL3 (MIP-1α, bead B9), IL-8 (bead A9), IL-6 (bead A7) were used. After 1 h of incubation, detection reagent was added to each sample. Following 2 h of incubation, samples were carefully washed twice prior flow cytometry acquisition on a BD FACS Canto II™ flow cytometer (BD Biosciences). Acquisition was adjusted by using the calibration bead procedure from the manufacturer and cytokine titration was quantified by using the standard curves and the Flow Cytometric Analysis Program Array software (FCAP-Array, Soft Flow) [62].

Secretion of IL-1β, IL-6 and CXCL2 in ER-Hoxb8 cells was measured by ELISA (Quantikine kits, R&D Systems, Minneapolis, MN, USA) according to the manufacturer’s instructions. Cell cytotoxicity was verified by using LDH (CytoTox 96 Non-Radioactive Cytotoxicity Assay, G1780; Promega), according to the manufacturer’s instructions.

### 4.10. Degranulation Analysis by Flow Cytometry

Degranulation was determined by the quantification of the expression at the plasma membrane of specific CD markers for azurophil granules (anti-CD63-PE, clone NVG, BD Biosciences), secondary granules (anti-CD18-BV421, clone C71/16, BD Biosciences), tertiary granules (anti-CD11b-APC, clone M1/70, ThermoFischer Scientific) and secretory vesicles (anti-CD14-PE, clone Sa2-8, ThermoFischer Scientific) using flow cytometry (FACS Canto II^TM^, BD Biosciences). IgG2a-PE (clone ebr2a, ThermoFischer Scientific), IgG2a-PerCP-eFluor^®^710 (clone ebr2a, ThermoFischer Scientific), IgG1-APC (cloneX40, BD Biosciences), IgG2a-BV421 (clone R35-38, BD Biosciences) and IgG1-BV510 (clone R3-R4, BD Biosciences) were used as negative isotype controls. The following antibodies were used in single dye staining to set up compensations: Ly6C/6G for PerCP-eFluor^®^710, CD11b for APC, CD16/32 for BV421 and CD45 for BV510. Data analysis were performed by recording the Median Fluorescence Intensity (MFI) for each CD markers with BD FACSDiva software (BD Biosciences) on the gated population of granulocytes (FSC-A vs. SSC-A), single (SSC-A vs. SSC-H) and living cells (negative for LIVE/DEAD^TM^ Fixable Near-IR Dead Cell Stain). No less than 10,000 single living cells were recorded *per* staining condition. The relative translocation of CD markers at the plasma membrane was determined by calculating the Staining Index (SI) of their specific CD markers based on the formula (MFI of CD marker—MFI of its isotype control)/(2 × rSD of its isotype control) [63]. For each stimulation time-point (2, 4, 6 and 12 h), a ratio between LPS stimulated and non-stimulated cells (SI+LPS/SI-LPS) was calculated for both WT differentiated and S100A9^−/−^ differentiated Hoxb8 cells.

### 4.11. Statistics

Statistical analyses were performed using GraphPad Prism software (GraphPad Software, La Jolla, CA, USA). Time-course results with multiple stimulatory conditions were analyzed with a two-way ANOVA analysis followed by a Tuckey multiple comparison test to compare the effect of stimulation between WT and S100A9^−/−^ at each time point. For two-group comparison, normality and homogeneity of variances were ascertained, as determined by Kolmogorov–Smirnov test and F-test respectively, and then Student’s *t*-test analyses were performed. Otherwise, Mann–Whitney tests were used for two-group comparisons. For all statistical tests, *p* ˂ 0.05 was considered statistically significant. * = *p* < 0.05; ** = *p* < 0.01; *** = *p* < 0.001; **** = *p* < 0.0001.

## Figures and Tables

**Figure 1 ijms-22-08845-f001:**
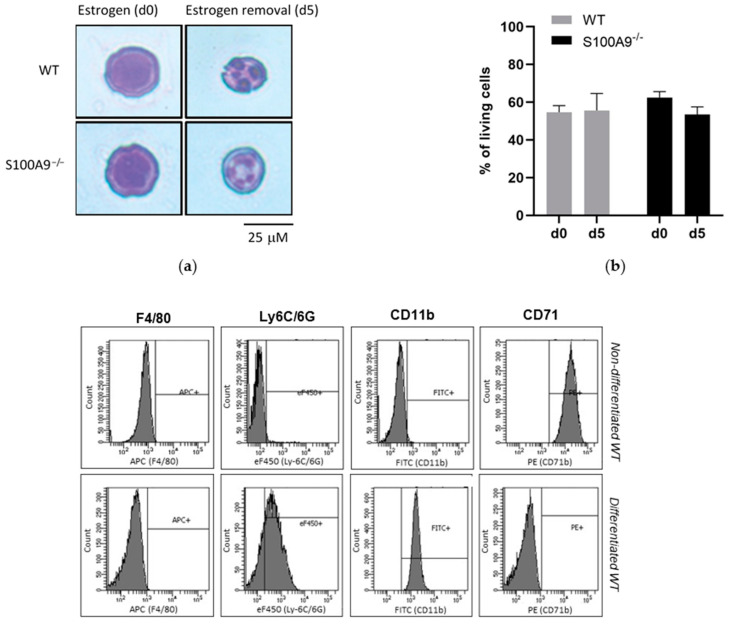
Morphological and phenotypic changes in wild type (WT) and S100A9^−/−^ Hoxb8 cells after differentiation. (**a**) Morphology of Hoxb8 cells after May–Grünwald Giemsa staining. Representative microscopic images of three independent experiments Hoxb8 before (day 0: d0) and after removal of estrogen (day 5: d5) for the induction of differentiation in the presence of 2% stem cell factor, scale bar: 25 μM. (**b**) Viability of Hoxb8 cells was determined by 7-AAD staining. 10,000 events were analyzed by flow cytometry and living cells were determined as percentage of 7-AAD-negatively labelled cells on the entire population of both WT and S100A9^−/−^ Hoxb8 cells. Data are presented as mean of % ± standard error of mean (SEM) of three independent experiments. (**c**) The neutrophilic differentiation of ER-Hoxb8 progenitors. F4/80, Ly6C/6G, CD11b and CD71 surface expression on WT Hoxb8 cells are showed (histograms) before and after differentiation. Surface marker expression was analyzed by flow cytometry using specific fluorescently labeled antibodies: anti-mouse F4/80 (macrophage marker), anti-mouse Ly6C/6G (or Gr-1, mouse granulocyte antigen), anti-mouse CD11b (myeloid differentiation antigen) and anti-mouse CD71 (immature myeloid antigen, [24]) antibodies. Percentage of WT and S100A9^−/−^ Hoxb8 cells expressing Ly6C/6G, CD11b and CD71 before and after differentiation. Data are expressed as mean ± SEM of three independent experiments. (**d**) S100A8 and S100A9 expression in Hoxb8 cells. Non-differentiated (d0) and differentiated (d5) Hoxb8 cells were lysed and RNA was extracted. *S100a8 and S100a9* mRNA transcripts were quantified using qPCR. Data are expressed as mean ± SEM of three independent experiments and were normalized to three reference genes *Actb*, *B2m* and *Gusb* [25]. * = *p* < 0.05; ** = *p* < 0.01; *** = *p* < 0.001; **** = *p* < 0.0001. On the right side, S100A8 and S100A9 proteins, in non-differentiated (d0) and differentiated (d5) Hoxb8 cells, were detected by Western Blot. β-actin was used as a loading control. This Western Blot is representative of three independent experiments.

**Figure 2 ijms-22-08845-f002:**
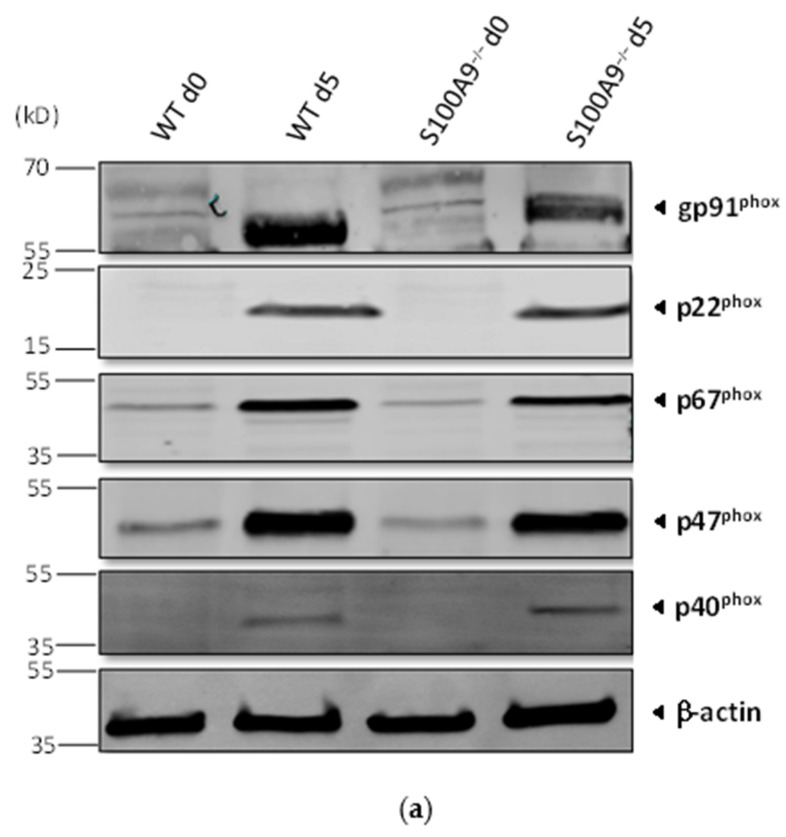
Nicotinamide adenine dinucleotide phosphate (NADPH) oxidase state in WT and S100A9^−/−^ differentiated Hoxb8 cells. (**a**) Expression of the NADPH oxidase subunits. gp91^phox^, p67^phox^, p47^phox^, p40^phox^ and p22^phox^ expression in non-differentiated (d0) and differentiated (d5) WT or S100A9^−/−^ Hoxb8 cells was determined by Western Blot. A representative Western Blot of three independent experiments is shown. (**b**) reactive oxygen species (ROS) production stimulated by N-Formylmethionyl-leucyl-phenylalanine (fMLF), phorbolmyristate acetate (PMA) or lipopolysaccharide (LPS). Differentiated WT and S100A9^−/−^ Hoxb8 cells were stimulated with fMLF (100 nM), PMA (100 nM) or LPS (100 ng/mL). Total ROS production was measured on a plate reader by real-time chemiluminescence. The results represent fold induction of ROS production of S100A9^−/−^ Hoxb8 cells compared to S100A9^−/−^ Hoxb8 cells and are expressed as mean ± SEM. Inserts: representative traces showing ROS production in WT Hoxb8 cells stimulated or not with fMLF, PMA or LPS. Significantly different from control * = *p* < 0.05, ** = *p* < 0.01.

**Figure 3 ijms-22-08845-f003:**
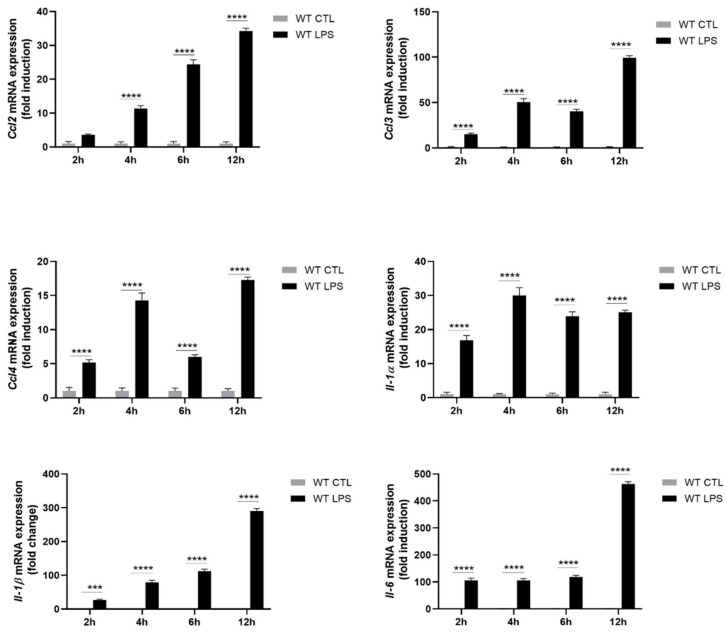
Expression of cytokines in WT differentiated Hoxb8 cells stimulated with LPS. Hoxb8 neutrophils were stimulated for 2, 4, 6 and 12 h without or with 100 ng/mL of LPS. The expression of *Ccl2*, *Ccl3*, *Ccl4*, *Il-1α*, *Il-1β*, *Il-6*, *Tnf-α* and *Cxcl2* mRNAs was assessed by qPCR. Data normalization was performed using three reference genes (*Actβ*, *B2m* and *Gusb*) and expressed as fold induction compared to non-stimulated control. Results are presented as mean ± SEM of three independent experiments. *** = *p* < 0.001; **** = *p* < 0.0001.

**Figure 4 ijms-22-08845-f004:**
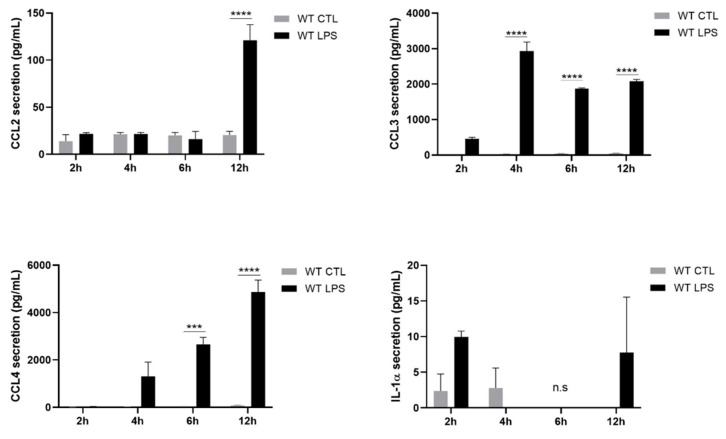
Secretion of cytokines in WT differentiated Hoxb8 cells stimulated with LPS. Differentiated Hoxb8 neutrophils were stimulated for 2, 4, 6, 12 h with 100 ng/mL of LPS. C-C motif chemokine ligand (CCL)-2, CCL3, CCL4, interleukin (IL)-1α, and tumor necrosis factor alpha (TNF-α) secretion was measured by cytokine cytometric bead array (CBA) analysis and IL-1β, IL-6 and C-X-C motif chemokine ligand 2 (CXCL2) secretion was quantified by ELISA. Results are presented as mean ± SEM of three independent experiments. * = *p* < 0.05; ** = *p* < 0.01; *** = *p* < 0.001; **** = *p* < 0.0001.

**Figure 5 ijms-22-08845-f005:**
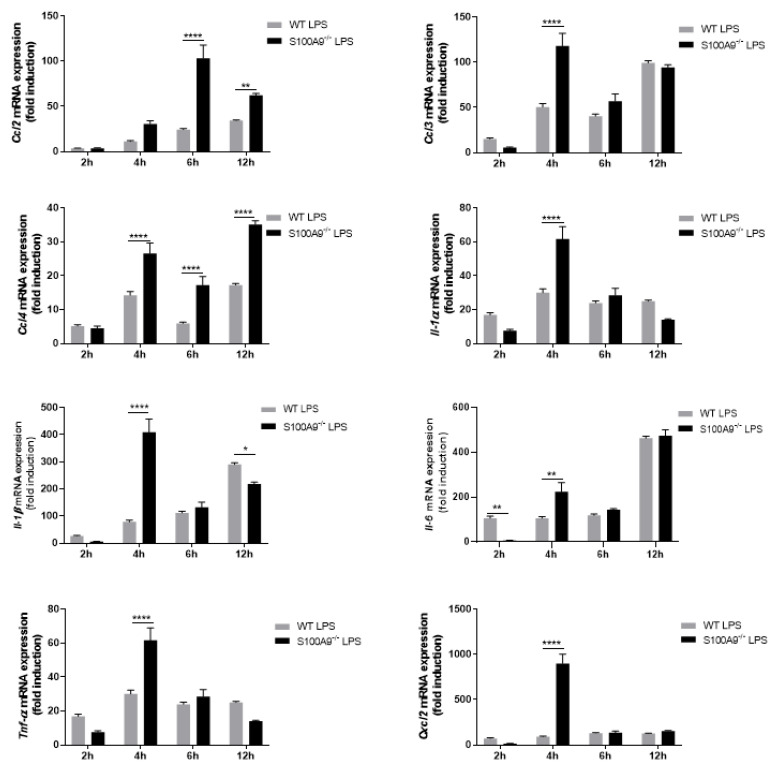
Effect of S100A9^−/−^ on LPS-induced cytokine mRNA expression in differentiated Hoxb8 cells. Differentiated WT and S100A9^−/−^ Hoxb8 cells were stimulated with 100 ng/mL LPS for 2, 4, 6 and, 12 h. Expression of *Ccl2*, *Ccl3*, *Ccl4*, *Il-1α*, *Il-1β*, *Il-6*, *Tnf-α* and *Cxcl2* mRNAs was assessed by qPCR. Data normalization was performed using three reference genes (*Actβ*, *B2m* and *Gusb*) and expressed as fold induction compared to the non-stimulated control for both WT and S100A9^−/−^. Results are shown as mean ± SEM of three independent experiments. * = *p* < 0.05; ** = *p* < 0.01; **** = *p* < 0.0001.

**Figure 6 ijms-22-08845-f006:**
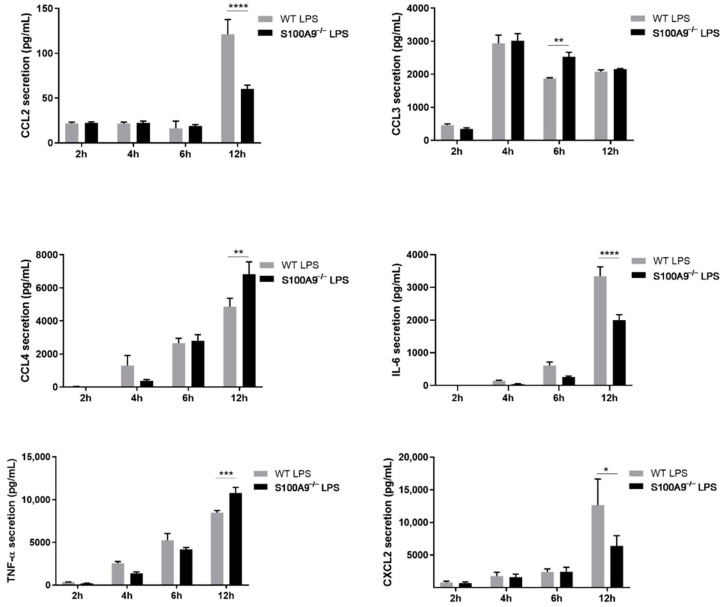
Effect of S100A9^−/−^ on LPS-induced cytokine secretion in differentiated Hoxb8 cells. Differentiated WT or S100A9^−/−^ cells were stimulated with 100 ng/mL of LPS for 2, 4, 6, 12 h. CCL2, CCL3, CCL4, and TNF-α secretion was measured by CBA analysis and IL-6 and CXCL2 secretion was quantified by ELISA. Results are presented as mean ± SEM of five independent experiments. * = *p* < 0.05; ** = *p* < 0.01; *** = *p* < 0.001; **** = *p* < 0.0001.

**Figure 7 ijms-22-08845-f007:**
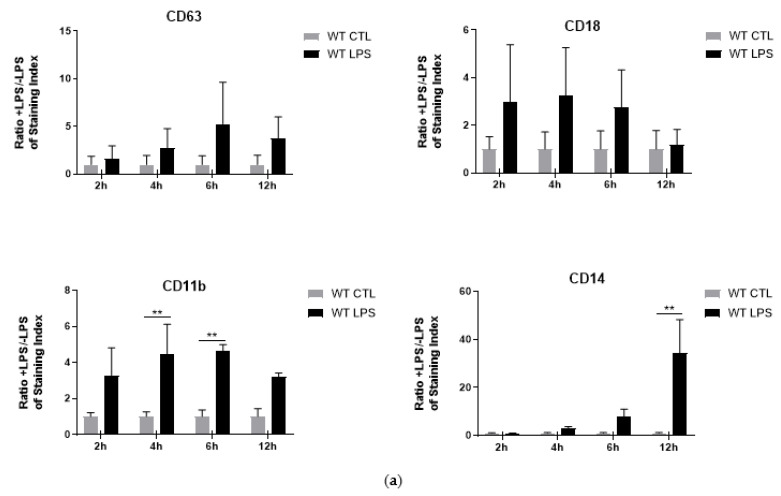
Effect of absence of the S100A9 protein in the process of degranulation. (**a**) Differentiated WT Hoxb8 cells were treated with 100 ng/mL LPS for 2, 4, 6 and 12 h. Translocation to the plasma membrane of selected granule markers (CD63 for azurophil granules, CD18 for specific granules, CD11b for gelatinase granules and CD14 for secretory vesicles) was assessed by flow cytometry. Results are expressed as a ratio of staining index (SI) of LPS-stimulated (+LPS) and non-stimulated (−LPS) cells at the same time point ± SEM of at least three independent experiments. Significantly different from non-stimulated control: ** = *p* < 0.01. (**b**) Differentiated WT or S100A9^−/−^ cells were stimulated with 100 ng/mL of LPS for 2, 4, 6, 12 h. Translocation to the plasma membrane of selected granule markers was assessed by flow cytometry. Results are expressed as a ratio of staining index (SI) of LPS-stimulated (+LPS) and non-stimulated (−LPS) cells at the same time point ± SEM of at least three independent experiments. Significantly different from non-stimulated control. ** = *p* < 0.01.

**Figure 8 ijms-22-08845-f008:**
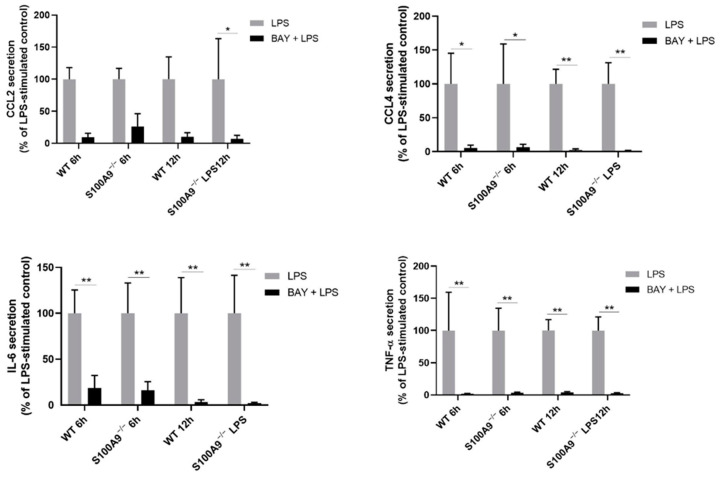
Relationship between NF-κB activation with S100A9-mediated cytokine secretion. Differentiated WT and S100A9^−/−^ Hoxb8 cells were stimulated with 100 ng/mL of LPS for 2, 4, 6 and 12 h after pre-treatment for 30 min with 10 µM NF-κB inhibitor BAY 11-7082. CCL2, IL-6, CCL4 and TNF-α secretion was measured by ELISA and expressed as percentage of LPS-stimulated samples. The results were presented as mean ± SEM of five independent experiments. * = *p* < 0.05; ** = *p* < 0.01.

**Figure 9 ijms-22-08845-f009:**
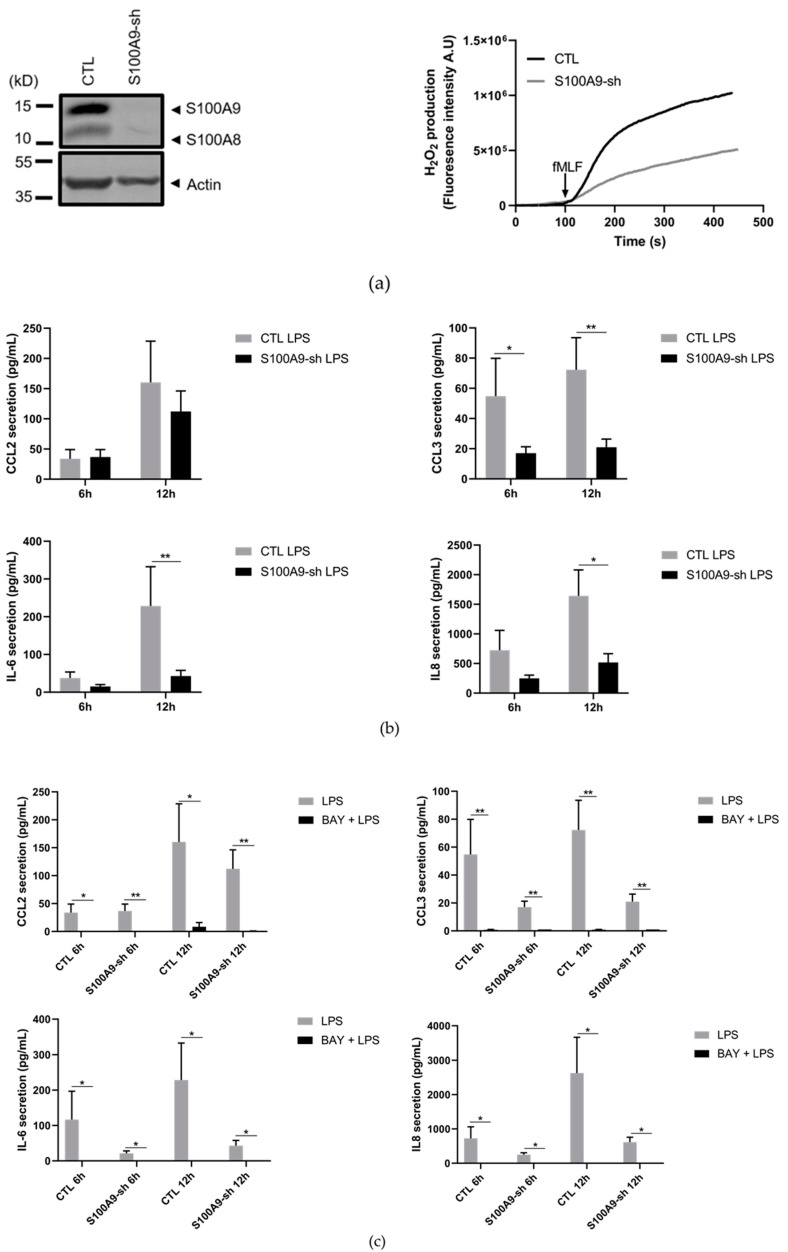
(**a**) Efficiency of S100A9 knock-down. Expression of S100A8 and S100A9 (left panel) in control and S100A9-shRNA differentiated HL-60 cells was determined by Western Blot. A representative Western Blot of three independent experiments is shown. ROS production (right panel) was measured in control and S100A9-shRNA differentiated HL-60 cells stimulated with fMLF (100 nM). A representative trace showing ROS production in control cells and S100A9-shRNA cells is shown. (**b**) Effect of S100A9 knock-down on LPS-induced cytokine secretion in differentiated HL-60 cells. Differentiated HL-60 cells (control and S100A9-shRNAs) were stimulated with 100 ng/mL of LPS for 6 and 12 h. CCL2, CCL3, IL-6, and IL-8 secretion was measured by CBA analysis. Results are presented as mean ± SEM of five independent experiments. * = *p* < 0.05; ** = *p* < 0.01. (**c**) Effect of inhibition of NF-κB activation in S100A9-mediated cytokine secretion. HL-60 cells (control and S100A9-shRNAs) were stimulated with 100 ng/mL of LPS for 6 and 12 h after pre-treatment for 30 min with 10 µM NF-κB inhibitor BAY 11-7082. CCL2, CCL3, IL-6, and IL-8 secretion was measured by CBA analysis. The results were presented as mean ± SEM of five independent experiments. * = *p* < 0.05; ** = *p* < 0.01.

**Table 1 ijms-22-08845-t001:** Primer sequences used for quantitative real-time PCR.

Gene	Forward (5′–3′)	Reverse (5′–3′)
*S100a8*	GTCCTCAGTTTGTGCAGAATA	CACCATCGCAAGGAACTC
*S100a9*	TTTAGCTTGAAGAGCAAGAAG	TGTCCTTCCTTCCTAGAGTATTG
*Ccl2*	ATCATCCCTGCGAGCCTATCCT	GACCTTTTTTGGCTAAACGCTTTC
*Ccl3*	ACTGCCTGCTGCTTCTCCTACA	ATGACACCTGGCTGGGAGCAAA
*Ccl4*	ACCCTCCCACTTCCTGCTGTTT	CTGTCTGCCTCTTTTGGTCAGG
*Il-1α*	ACGGCTGAGTTTCAGTGAGACC	CACTCTGGTAGGTGTAAGGTGC
*Il-1β*	TGGACCTTCCAGGATGAGGACA	GTTCATCTCGGAGCCTGTAGTG
*Il-6*	TACCACTTCACAAGTCGGAGGC	CTGCAAGTGCATCATCGTTGTTC
*Tnf-α*	GGTGCCTATGTCTCAGCCTCTT	GCCATAGAACTGATGAGAGGGAG
*Cxcl2*	CCAAGGGTTGACTTCAAGAAC	TGAGAGTGGCTATGACTTCTG
*Actβ*	AGCCTTCCTTCTTGGGTATG	AGCACTGTGTTGGCATAGA
*18s*	AGCGAGTGATCACCATCAT	CCAGAACCTGGCTGTACTT
*B2m*	TCACACTGAATTCACCCCCAC	TGATCACATGTCTCGATCCCAG

## Data Availability

The data presented in this study are available on request from the corresponding author.

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
