# Peer review of "Role of S100A8/A9 for Cytokine Secretion, Revealed in Neutrophils Derived from ER-Hoxb8 Progenitors"

_ijms, 2021, doi:10.3390/ijms22168845_

Round 1

Reviewer 1 Report

The manuscript describes the role of intracellular S100A8/A9 in cytokine secretion of neutrophils by using neutrophils derived from wild type and S100A9-/- Hoxb8 immortalized myeloid progenitors (wild type and S100A9-/- Hoxb8 neutrophil-like cells).  Based on the results, the authors claimed that Hoxb8 neutrophil-like cells may be a good model for genetically manipulated murine neutrophils, and they suggested that intracellular S100A8/A9 is involved in LPS-induced cytokine release in neutrophils.  Because wild type Hoxb8 neutrophil-like cells showed granule exocytosis dissimilar to neutrophils, the authors should confirm their findings by using HL-60 cells.  Although some readers may be interested in their findings, there are some other concerns in the manuscript as follows. 

(1) Since the authors did not examine whether or not wild type Hoxb8 neutrophil-like cells can release S100A8/A9, the possibility remains that extracellular S100A8/A9 may also exert the functions described in the manuscript.

(2) In Fig. 1 (c), the authors should show the dot plot data for flow cytometry and whether or not S100A9-/- Hoxb8 neutrophil-like cells contain monocytes/macrophages and other types of cells. 

(3) In Fig. 1 (d), S100A8 mRNA but not S100A8 protein is detected.  The authors should discuss about the possible cause for this.

(4) In Fig.2 (b), the data for LPS and PMA-induced ROS production are wrongly located. 

(5) There are grammatical errors in the manuscript, and therefore the authors are asked to consult with an expert in editing English.

Author Response

1- Since the authors did not examine whether or not wild type Hoxb8 neutrophil-like cells can release S100A8/A9, the possibility remains that extracellular S100A8/A9 may also exert the functions described in the manuscript.

Thank you for this important comment. It has previously been reported that LPS was able to induce the secretion of S100A8/A9 by myeloid cells (Suryono et al., J. Periodontol. 2003, 74:1719-1724). Therefore, it is highly likely that S100A8/A9 are present in the extracellular environment of WT Hoxb8 cells. However, in the absence of S100A9 (S100A9-/- Hoxb8 cells), cytokine RNA expression was dysregulated (up- or down-regulation, see Figure 5) and a similar observation was reported for the level of cytokine secretion (Figure 6). Since in S100A9-/- Hoxb8 cells, no S100A8 and S100A9 proteins are found in the cytosol, we can assume that no secretion of S100A8/A9 occurs and the effects observed are the consequence of the absence of intracellular S100A9.

2- In Fig. 1 (c), the authors should show the dot plot data for flow cytometry and whether or not S100A9-/- Hoxb8 neutrophil-like cells contain monocytes/macrophages and other types of cells. 

As requested by the reviewers, we have included data for flow cytometry. In the revised manuscript, histograms representing F4/80, Ly6C/6G, CD11b and CD71 surface expression on WT Hoxb8 cells are showed before and after differentiation (Figure 1c; page 2, lines 90-90; page 4, lines 175-181). As specified in the manuscript (page 2 lines 91-94), the rate of differentiation of Hoxb8 cells (WT and S100A9-/-), obtained from our experimental conditions are in accordance with those previously reported (> 70% of neutrophils) (Wang et al., Nat. Methods 2006, 3:287-293; Chang et al., BMC Cell Biol. 2006, 7:11; Lee et al., J. Leukoc. Biol. 2013, 94:585-594). However, and as previously observed in the other studies using Hoxb8 cells (e.g. Saul et al., J. Leukoc. Biol. 2019, 106:1101-1115), contaminating cells are present in our WT and S100A9-/- Hoxb8 cell preparation (see Figure 1c). In this sense, an adapted protocol to eliminate the fraction of Gr-1 (Ly-6C/6G) negative cells by flow cytometry needs to be implemented in order to reach, in the future, a rate of 99% of neutrophils (page 17, lines 695-697 in the revised manuscript). Because we are aware of this issue, our results have been validated on differentiated HL-60 cells since these cells are not only human neutrophil-like cells but also a well-characterized model without contaminating cells.

3- In Fig. 1 (d), S100A8 mRNA but not S100A8 protein is detected.  The authors should discuss about the possible cause for this.

Thank you for pointing this out. Indeed, this has not been discussed. The fact that S100A8 mRNA but not S100A8 protein is detected in the S100A9-/- myeloid cells has previously been reported (Manitz et al., Mol. Cell Biol. 2003, 23:1034-1043 and Hobbs et al., Mol. Cell Biol. 2003, 23:2564-2576). The absence of S100A8 protein has been proposed to be caused by an inefficient translation of S100A8 mRNA but the predominant hypothesis is that the absence of S100A9 protein (the partner of S100A8) leads to an instability of S100A8 protein (Hobbs et al., Mol. Cell Biol. 2003, 23:2564-2576). A sentence has been added in the manuscript (page 3, lines 104-107) on the possible cause of the absence of S100A8 protein.

4- In Fig.2 (b), the data for LPS and PMA-induced ROS production are wrongly located. 

Again, thank you for pointing this out. Indeed, the graph corresponding to LPS and PMA-induced ROS production have been inverted. This was corrected in the revised manuscript (Figure 2b).

5- There are grammatical errors in the manuscript, and therefore the authors are asked to consult with an expert in editing English.

As requested by the reviewer, the manuscript has been carefully proofread by an appropriate person and correction of grammatical errors have been made.

Reviewer 2 Report

The aim is stated clear. The authors stated clearly what study found and how they did it.

The title is informative and relevant.

The references are relevant and recent. The cited sources are referenced correctly. Appropriate and key studies are included.

The introduction reveals what is already known about this topic. The research question also justified given what is already known about the topic.

The study methods are valid and reliable. There are enough details provided in order to replicate the study.

The data is presented in an appropriate way. Statistically significant results are clear. It is clear which results are with practical meaning. Results are discussed from different angles and placed into context without being overinterpreted.

The conclusions answer the aim of the study. The conclusions are supported by references and own results.

The limitations of the study are not fatal, but they are opportunities to inform future research.

Specific comments on weaknesses of the article and what could be improved:

Major points  - none

Minor points

  1. You stated that "However, our data do not 342
    allow to conclude on the role of S100A9 in the transcriptional response, which supports 343
    cytokine synthesis." But ould you please speculate on the clinical aspects and implications of the results.

Author Response

1- You stated that "However, our data do not allow to conclude on the role of S100A9 in the transcriptional response, which supports cytokine synthesis". But could you please speculate on the clinical aspects and implications of the results.

As requested by the reviewer, the clinical aspects on the role of S100A9 in the cytokine transcriptional regulation has been discussed in the revised manuscript (page 16, lines 660-669).

Round 2

Reviewer 1 Report

The authors responded to almost all my comments except for comment (1), and they modified the manuscript accordingly.

Based on previous publications, the authors speculate that S100A8/A9 are present in the extracellular environment of WT Hoxb8 cells.  Since in S100A9-/- Hoxb8 cells no S100A8 and S100A9 proteins are found in the cytosol, no S100A8 and S100A9 proteins should be found in the extracellular environment of S100A9-/- Hoxb8 cells.   Consequently, they are allowed to conclude that S100A8 and S100A9 proteins are involved in cytokine secretion, but they are not allowed to conclude that intracellular S100A8 and S100A9 proteins are involved in cytokine secretion.  They should examine the effect of the addition of exogenous S100A8 and S100A9 proteins in S100A9-/- Hoxb8 cells.

Author Response

Based on previous publications, the authors speculate that S100A8/A9 are present in the extracellular environment of WT Hoxb8 cells. Since in S100A9-/- Hoxb8 cells no S100A8 and S100A9 proteins are found in the cytosol, no S100A8 and S100A9 proteins should be found in the extracellular environment of S100A9-/- Hoxb8 cells. Consequently, they are allowed to conclude that S100A8 and S100A9 proteins are involved in cytokine secretion, but they are not allowed to conclude that intracellular S100A8 and S100A9 proteins are involved in cytokine secretion.  They should examine the effect of the addition of exogenous S100A8 and S100A9 proteins in S100A9-/- Hoxb8 cells.

We thank the reviewer for this additional comment.

In the first revision, we had addressed the reviewer’s point by answering affirmatively on the possibility that wild type Hoxb8 neutrophil-like cells can release S100A8/A9 upon LPS stimulation. Subsequently, the reviewer now proposes to stimulate S100A9-/- Hoxb8 cells by exogenous S100A8 and S100A9 in order to determine whether cytokine secretion is regulated by extracellular S100A8/A9 or intracellular S100A8/A9.

In this context, it is important to specify that our previous work (Schenten et al., Front. Immunol. 2018, 9:447) provided evidence that exogenous S100A8/A9 was able to induce cytokine secretion in differentiated HL-60 cells probably through the TLR4 signalling pathway similarly to stimulation with LPS.

We are strongly convinced that both extracellular S100A8/A9 and intracellular S100A8/A9 are involved in the regulation of cytokine secretion since inhibition of S100A9 leads to a decrease in cytokine secretion and addition of extracellular S100A8/A9 or LPS leads to an increase of cytokine secretion via the TLR4 signalling pathway.

However, we agree with the reviewer that further experiments are required to determine more precisely the role of intracellular versus extracellular S100A8/A9; this constitutes a logical continuation for future research projects on this topic (S100A8/A9 and cytokine secretion).

As mentioned by the reviewer it will be interesting to stimulate S100A9-/- Hoxb8 cells but also S100A9-shRNA HL-60 cells by exogenous S100A8/A9. However, the results of this type of experiments will not allow to distinguish the role of intracellular S100A8/A9 vs. extracellular S100A8/A9. Indeed, if cytokine secretion is observed in the absence of intracellular S100A8/A9, this could lead to the assumption that intracellular S100A8/A9 is not involved in cytokine secretion induced by S100A8/A9 stimulation. In the case where cytokine secretion is affected by the inhibition of intracellular S100A8/A9, the conclusion could be that intracellular S100A8/A9 is involved in the regulation of signalling pathways leading to cytokine production and secretion (as seen with LPS).

Prior to S100A8/A9 stimulation of cells, the concentration of secreted S100A8/A9 would need to be determined in order to stimulate the cells with similar concentrations. Furthermore, the stimulation of cells will be performed with recombinant proteins and this could largely modify the response compared to “physiological” S100A8/A9 proteins. In parallel, experiments need to be conducted with S100A8/A9 blocking antibodies in order to inhibit extracellular activity of S100A8/A9. Moreover, it also appears important to define whether intracellular S100A8/A9 are similarly involved in signalling pathways activated by S100A8/A9 and LPS.

Again, we thank the reviewer for this relevant and important point. As previously mentioned, although the different data rather tend to underline a role of intracellular S100A8/A9, we agree that the exact involvement of extracellular and intracellular S100A8/A9 needs to more deeply investigated. As outlined above, several important parameters will need to be established and fine-tuned upfront and this is currently in the planning as part of a new research project and this would, to our mind, not be possible within a revision process. However, to emphasize this important point, we added a sentence to the discussion of the revised manuscript (page 17, lines 702-705).

Round 3

Reviewer 1 Report

The authors responded to my comment, and they added the sentences to the manuscript accordingly.  However, they did not delete the word "intracellular”, for instance, from the title.  Since the present data are not sufficient to conclude the role of intracellular S100A8/A9,  they should modify the manuscript by deleting the word wherever possible.

Author Response

The authors responded to my comment, and they added the sentences to the manuscript accordingly.  However, they did not delete the word "intracellular”, for instance, from the title.  Since the present data are not sufficient to conclude the role of intracellular S100A8/A9, they should modify the manuscript by deleting the word wherever possible.

We thank the reviewer for this additional comment.

We have deleted “intracellular” from the title in the revised manuscript. Moreover, this word has been deleted or “the absence of intracellular S100A8/A9” has been specified in an appropriate manner in the revised manuscript (page 1, lines 20 and 29; page 2, line 73; page 5, line 203; page 6, lines 264 and 265; page 10, line 393: page 15, line 616; page 17, line 700).